# How Well Do WGANs Estimate the Wasserstein Metric?

## Abstract

Generative modelling is often cast as minimizing a similarity measure between a data distribution and a model distribution. Recently, a popular choice for the similarity measure has been the Wasserstein metric, which can be expressed in the Kantorovich duality formulation as the optimum difference of the expected values of a potential function under the real data distribution and the model hypothesis. In practice, the potential is approximated with a neural network and is called the discriminator. Duality constraints on the function class of the discriminator are enforced approximately, and the expectations are estimated from samples. This gives at least three sources of errors: the approximated discriminator and constraints, the estimation of the expectation value, and the optimization required to find the optimal potential. In this work, we study how well the methods, that are used in generative adversarial networks to approximate the Wasserstein metric, perform. We consider, in particular, the $c$-transform formulation, which eliminates the need to enforce the constraints explicitly. We demonstrate that the $c$-transform allows for a more accurate estimation of the true Wasserstein metric from samples, but surprisingly, does not perform the best in the generative setting.

## 1 Introduction

Recently, optimal transport (OT) has become increasingly prevalent in computer vision and machine learning, as it allows for robust comparison of structured data that can be cast as probability measures, e.g., images, point clouds and empirical distributions (Rubner et al., 2000; Lai & Zhao, 2014), or more general measures (Gangbo et al., 2019). Key properties of OT include its non-singular behavior, when comparing measures with disjoint supports, and the fact that OT inspired objectives can be seen as lifting similarity measures between samples to similarity measures between probability measures. This is in stark contrast to the more traditional information theoretical divergences, which rely on only comparing the difference in mass assignment. Additionally, when the *cost function* $c$ is related to a distance function $d$ by $c(x, y) = d^p(x, y), p \geq 1$, the OT formulation defines the so called *Wasserstein metric*, which is a distance on the space of probability measures, i.e. a symmetric and positive definite function that satisfies the triangle inequality. Despite its success, scaling OT to big data applications has not been without challenges, since it suffers from the curse of dimensionality (Dudley, 1969; Weed & Bach, 2017). However, significant computational advancements have been made recently, for which a summary is given by Peyré et al. (2019). Notably, the *entropic regularization* of OT introduced by Cuturi (2013) preserves the geometrical structure endowed by the Wasserstein spaces and provides an efficient way to approximate optimal transport between measures.

**Generative modelling**, where *generators* are trained for sampling from given data distributions, is a popular application of OT. In this field, *generative adversarial networks* (GANs) by Goodfellow et al. (2014) have attracted substantial interest, particularly due to their success in generating photo-realistic images (Karras et al., 2017). The original GAN formulation minimizes the *Jensen-Shannon divergence* between a model distribution and the data distribution, which suffers from unstable training. *Wasserstein GANs* (WGANs) (Arjovsky et al., 2017) minimize the 1-Wasserstein distance over the $l^2$-metric, instead, resulting in more robust training.

The main challenge in WGANs is estimating the Wasserstein metric, consisting of estimating expected values of the *discriminator* from samples (drawn from a model distribution and a given data distribution), and optimizing the discriminator to maximize an expression of these expected

values. The discriminators are functions from the sample space to the real line, that have different interpretations in different GAN variations. The main technical issue is that the discriminators have to satisfy specific conditions, such as being 1-Lipschitz in the 1-Wasserstein case. In the original paper, this was enforced by clipping the weights of the discriminator to lie inside some small box, which, however, proved to be inefficient. The *Gradient penalty WGAN* (WGAN-GP) (Gulrajani et al., 2017) was more successful at this, by enforcing the constraint through a gradient norm penalization. Another notable improvement was given by the *consistency term WGAN* (CT-WGAN) (Wei et al., 2018), which penalizes diverging from 1-Lipschitzness directly. Other derivative work of the WGAN include different OT inspired similarity measures between distributions, such as the *sliced Wasserstein distance* (Deshpande et al., 2018), the *Sinkhorn divergence* (Genevay et al., 2018) and the *Wasserstein divergence* (Wu et al., 2018). Another line of work studies how to incorporate more general ground cost functions than the $l^2$-metric (Adler & Lunz, 2018; Mallasto et al., 2019; Dukler et al., 2019).

Recent works have studied the convergence of estimates of the Wasserstein distance between two probability distributions, both in the case of continuous (Klein et al., 2017) and finite (Sommerfeld & Munk, 2018; Sommerfeld, 2017) sample spaces. The decay rate of the approximation error of estimating the true distance with minibatches of size $N$ is of order $O(N^{-1/d})$ for the Wasserstein distances, where $d$ is the dimension of the sample space (Weed & Bach, 2017). Entropic regularized optimal transport has more favorable sample complexity of order $O(1/\sqrt{N})$ for suitable choices of regularization strength (see Genevay et al. 2019 and also Feydy et al. 2018; Mena & Weed 2019). For this reason, entropic relaxation of the 1-Wasserstein distance is also considered in this work.

**Contribution.** In this work, we study the efficiency and stability of computing the Wasserstein metric through its dual formulation under different schemes presented in the WGAN literature. We present a detailed discussion on how the different approaches arise and differ from each other qualitatively, and finally measure the differences in performance quantitatively. This is done by quantifying how much the estimates differ from accurately computed ground truth values between subsets of commonly used datasets. Finally, we measure how well the distance is approximated during the training of a generative model. This results in a surprising observation; the method best approximating the Wasserstein distance does not produce the best looking images in the generative setting.

## 2 OPTIMAL TRANSPORT

In this section, we recall essential formulations of optimal transport to fix notation.

**Probabilistic Notions.** Let $(\mathcal{X}, d_{\mathcal{X}})$ and $(\mathcal{Y}, d_{\mathcal{Y}})$ be Polish spaces, i.e., complete and separable metric spaces, denote by $\mathcal{P}(\mathcal{X})$ the set of probability measures on $\mathcal{X}$, and let $f \colon \mathcal{X} \to \mathcal{Y}$ be a measurable map. Then, given a probability measure $\mu \in \mathcal{P}(\mathcal{X})$, we write $f_{\#}\mu$ for the *push-forward* of $\mu$ with respect to $f$, given by $f_{\#}\mu(A) = \mu(f^{-1}(A))$ for any measurable $A \subseteq \mathcal{Y}$. Intuitively speaking, if $\xi$ is a random variable with law $\mu$, then $f(\xi)$ has law $f_{\#}\mu$. Then, given $\nu \in \mathcal{P}(\mathcal{Y})$, we define

$$\mathrm{ADM}(\mu, \nu) = \{\gamma \in \mathcal{P}(\mathcal{X} \times \mathcal{Y}) | \ (\pi_1)_{\#}\gamma = \mu, \ (\pi_2)_{\#}\gamma = \nu\}, \tag{1}$$

where $\pi_i$ denotes the projection onto the $i^{\text{th}}$ marginal. An element $\gamma \in \mathrm{ADM}(\mu, \nu)$ is called an *admissible plan* and defines a joint probability between $\mu$ and $\nu$.

**Optimal Transport Problem.** Given a continuous and lower-bounded *cost function* $c \colon \mathcal{X} \times \mathcal{Y} \to \mathbb{R}$, the optimal transport problem between probability measures $\mu \in \mathcal{P}(\mathcal{X})$ and $\nu \in \mathcal{P}(\mathcal{Y})$ is defined as

$$\mathrm{OT}_c(\mu, \nu) := \min_{\gamma \in \mathrm{ADM}(\mu, \nu)} \mathbb{E}_{\gamma}[c], \tag{2}$$

where $\mathbb{E}_{\mu}[f] = \int_{\mathcal{X}} f(x) d\mu(x)$ is the expectation of a measurable function $f$ with respect to $\mu$.

Note that (2) defines a constrained linear program, and thus admits a *dual formulation*. From the perspective of WGANs, the dual is more important than the primal formulation, as it can be approximated using *discriminator* neural networks. Denote by $L^1(\mu) = \{f \colon \mathcal{X} \to \mathbb{R} \mid \mathbb{E}_{\mu}[f] < \infty\}$ the set of measurable functions of $\mu$ that have finite expectations under $\mu$, and by $\mathrm{ADM}(c)$ the set of *admissible pairs* $(\varphi, \psi)$ that satisfy

$$\varphi(x) + \psi(y) \leq c(x, y), \quad \forall (x, y) \in \mathcal{X} \times \mathcal{Y}, \quad \varphi \in L^1(\mu), \psi \in L^1(\nu). \tag{3}$$

Then, the following duality holds (Peyré et al., 2019, Sec. 4)

$$\mathrm{OT}_c(\mu, \nu) = \sup_{(\varphi, \psi) \in \mathrm{ADM}(c)} \{\mathbb{E}_{\mu}[\varphi] + \mathbb{E}_{\nu}[\psi]\}. \tag{4}$$

When the supremum is attained, the optimal $\varphi^*, \psi^*$ in (4) are called *Kantorovich potentials*, which, in particular, satisfy $\varphi^*(x) + \psi^*(y) = c(x, y)$ for any $(x, y) \in \mathrm{Supp}(\gamma^*)$, where $\gamma^*$ solves (2). Given $\varphi$, we can obtain an admissable $\psi$ satisfying (4) through the *c-transform* of $\varphi$,

$$\varphi^c : \mathcal{Y} \to \mathbb{R}, \quad y \mapsto \inf_{x \in \mathcal{X}} \left\{ c(x, y) - \varphi(x) \right\}, \tag{5}$$

so that $(\varphi, \varphi^c) \in \mathrm{ADM}(c)$ for any $\varphi \in L^1(\mu)$. Moreover, the Kantorovich potentials satisfy $\psi = \varphi^c$, and therefore (4) can be written as (Villani, 2008, Thm. 5.9)

$$\mathrm{OT}_c(\mu, \nu) = \max_{(\varphi, \varphi^c) \in \mathrm{ADM}(c)} \left\{ \mathbb{E}_\mu[\varphi] + \mathbb{E}_\nu[\varphi^c] \right\}. \tag{6}$$

In other words, the $\mathrm{ADM}(c)$ constraint can be enforced with the $c$-transform, and reduces the optimization in (6) to be carried out over a single function.

**Wasserstein Metric.** Consider the case when $\mathcal{X} = \mathcal{Y}$ and $c(x, y) = d_{\mathcal{X}}^p(x, y)$, $p \geq 1$, where we refer to $d_{\mathcal{X}}$ as the *ground metric*. Then, the optimal transport problem (2) defines the *p-Wasserstein metric* $W_p(\mu, \nu) := \mathrm{OT}_{d_{\mathcal{X}}^p}(\mu, \nu)^{\frac{1}{p}}$ on the space

$$\mathcal{P}_{d_{\mathcal{X}}}^p(X) = \left\{ \mu \in \mathcal{P}(X) \,\middle|\, \int d_{\mathcal{X}}^p(x_0, x) d\mu(x) < \infty \right\}, \quad \text{for some } x_0 \in \mathcal{X}, \tag{7}$$

of probability measures with finite $p$-moments. It can be shown that $(\mathcal{P}_{d_{\mathcal{X}}}^p(\mathcal{X}), W_p)$ forms a complete, separable metric space (Villani, 2008, Sec. 6, Thm 6.16).

**Entropy Relaxed Optimal Transport.** We can relax (2) by imposing entropic penalization introduced by Cuturi (2013). Recall the definition of the *Kullback-Leibler (KL) divergence* from $\nu$ to $\mu$ as

$$\mathrm{KL}(\mu \| \nu) = \int_{\mathcal{X}} \log \left( \frac{p_\mu}{p_\nu} \right) p_\mu d\chi, \tag{8}$$

where we assume that $\mu, \nu$ are absolutely continuous with respect to the Lebesgue measure $\chi$ on $\mathcal{X}$ with densities $p_\mu, p_\nu$, respectively. Using the KL-divergence as penalization, the *entropy relaxed optimal transport* is defined as

$$\mathrm{OT}_c^\epsilon(\mu, \nu) := \min_{\gamma \in \mathrm{ADM}(\mu, \nu)} \left\{ \mathbb{E}_\gamma[c] + \epsilon \mathrm{KL}(\gamma \| \mu \otimes \nu) \right\}, \tag{9}$$

where $\epsilon > 0$ defines the magnitude of the penalization, and $\mu \otimes \nu$ denotes the independent joint distribution of $\mu$ and $\nu$. We remark that when $\epsilon \to 0$, any minimizing sequence $(\gamma^\epsilon)_{\epsilon > 0}$ solving (9) converges to a minimizer of (2), and in particular, $\mathrm{OT}_c^\epsilon(\mu, \nu) \to \mathrm{OT}_c(\mu, \nu)$.

Analogously to (4), the entropy relaxed optimal transport admits the following dual formulation (Peyré et al., 2019; Feydy et al., 2018; Di Marino & Gerolin, 2019)

$$\mathrm{OT}_c^\epsilon(\mu, \nu) = \sup_{\varphi \in L^1(\mu), \psi \in L^1(\nu)} \left\{ \mathbb{E}_\mu[\varphi] + \mathbb{E}_\nu[\psi] - \epsilon \mathbb{E}_{\mu \otimes \nu} \left[ \exp \left( \frac{-c + (\varphi \oplus \psi)}{\epsilon} \right) - 1 \right] \right\}, \tag{10}$$

where $(\varphi \oplus \psi)(x, y) = \varphi(x) + \psi(y)$. In contrast to (4), this is an *unconstrained* optimization problem, where the entropic penalization can be seen as a smooth relaxation of the constraint.

As shown by Feydy et al. (2018); Di Marino & Gerolin (2019), a similar approach to (6) for computing the Kantorovich potentials can be taken in the entropic case, by generalizing the $c$-transform. Let $L_\epsilon^{\exp}(\mu) := \{ g : \mathcal{X} \to \mathbb{R} \mid \mathbb{E}_\mu[\exp(g/\epsilon)] < \infty \}$, and consider the $(c, \epsilon)$-transform of $\varphi \in L_\epsilon^{\exp}(\mu)$,

$$\varphi^{(c, \epsilon)}(y) = -\epsilon \log \left( \int_{\mathcal{X}} \exp \left( -\frac{c(x, y) - \varphi(x)}{\epsilon} \right) d\mu(x) \right). \tag{11}$$

As $\epsilon \to 0$, $\varphi^{(c, \epsilon)}(y) \to \varphi^c(y)$, making the $(c, \epsilon)$-transform consistent with the $c$-transform. Analogously to (6), one can show under mild assumptions on the cost $c$ (Di Marino & Gerolin, 2019), that

$$\mathrm{OT}_c^\epsilon(\mu, \nu) = \max_{\varphi \in L_\epsilon^{\exp}(\mu)} \left\{ \mathbb{E}_\mu[\varphi] + \mathbb{E}_\nu[\varphi^{(c, \epsilon)}] \right\}. \tag{12}$$

**Sinkhorn Divergence.** Since the functional $\mathrm{OT}_c^\epsilon$ fails to be positive-definite (e.g. $\mathrm{OT}_c^\epsilon(\mu, \mu) \neq 0$), it is convenient to introduce the $(p, \epsilon)$-*Sinkhorn divergence* $S_p^\epsilon$ with parameter $\epsilon > 0$, given by

$$S_p^\epsilon(\mu, \nu) = \mathrm{OT}_{d_{\mathcal{X}}^p}^\epsilon(\mu, \nu)^{\frac{1}{p}} - \frac{1}{2} \left( \mathrm{OT}_{d_{\mathcal{X}}^p}^\epsilon(\mu, \mu)^{\frac{1}{p}} + \mathrm{OT}_{d_{\mathcal{X}}^p}^\epsilon(\nu, \nu)^{\frac{1}{p}} \right), \tag{13}$$

where the terms $\mathrm{OT}_{d_{\mathcal{X}}^p}^\epsilon(\mu, \mu)^{\frac{1}{p}}$ and $\mathrm{OT}_{d_{\mathcal{X}}^p}^\epsilon(\nu, \nu)^{\frac{1}{p}}$ are added to avoid bias, as in general $\mathrm{OT}_{d_{\mathcal{X}}^p}^\epsilon(\mu, \mu)^{\frac{1}{p}} \neq 0$. The Sinkhorn divergence was introduced by Genevay et al. (2018), and has the following properties: (i) it metrizes weak convergence in the space of probability measures; (ii) it interpolates between *maximum mean discrepancy* (MMD), as $\epsilon \to \infty$, and the $p$-Wasserstein metric, as $\epsilon \to 0$. For more about the Sinkhorn divergence, see Feydy et al. (2018).

## 3 GENERATIVE ADVERSARIAL NETWORKS

*Generative Adversarial Networks* (GANs) are popular for learning to sample from distributions. The idea behind GANs can be summarized as follows: given a *source distribution* $\mu_s \in \mathcal{P}(\mathbb{R}^{n_s})$, we want to push it forward by a parametrized *generator* $g_{\omega'}\colon \mathbb{R}^{n_s} \to \mathbb{R}^{n_t}$, so that a chosen similarity measure $\rho$ between the pushforward distribution and the *target distribution* $\mu_t \in \mathcal{P}(\mathbb{R}^{n_t})$ is minimized. Usually the target distribution is only accessible in form of a dataset of samples, and one considers an 'empirical' version of the distributions. Note that $n_s \ll n_t$ is chosen, which is justified by the *manifold hypothesis*. This objective can be expressed as

$$\min_{\omega'} \rho((g_{\omega'})_\# \mu_s, \mu_t). \tag{14}$$

The similarity $\rho$ is commonly estimated with a *discriminator* $\varphi_\omega\colon \mathbb{R}^{n_t} \to \mathbb{R}$, parametrized by $\omega$, whose role will become apparent below.

**The vanilla GAN** (Goodfellow et al., 2014) minimizes an approximation to the *Jensen-Shannon (JS) divergence* between the push-forward and target, given by

$$\mathrm{JS}(\nu\|\mu) \approx \max_\omega \left\{ \mathbb{E}_{x\sim\mu} \left[\log(\varphi_\omega(x))\right] + \mathbb{E}_{y\sim\nu} \left[\log(1 - \varphi_\omega(y))\right] \right\}, \tag{15}$$

for probability measures $\mu$ and $\nu$. The discriminator $\varphi_\omega$ is restricted to take values between $0$ and $1$, assigning a probability to whether a point lies in $\mu$ or $\nu$. It can be shown (Goodfellow et al., 2014), that at optimality the JS-divergence is recovered in (15), if optimized over all possible functions. Substituting $\mu = \mu_t$ and $\nu = (g_{\omega'})_\# \mu_s$ in (15) yields the minimax objective for the vanilla GAN

$$\min_{\omega'} \max_\omega \left\{ \mathbb{E}_{x\sim\mu_t} \left[\log(\varphi_\omega(x))\right] + \mathbb{E}_{z\sim\mu_s} \left[\log(1 - \varphi_\omega(g_{\omega'}(z)))\right] \right\}. \tag{16}$$

As mentioned above, in practice one considers empirical versions of the distributions so that the expectations are replaced by sample averages.

**The Wasserstein GANs** (WGANs) (Arjovsky et al., 2017) minimize an approximation to the 1-Wasserstein metric over the $l^2$ ground metric, instead. The reason why the $p = 1$ Wasserstein case is considered is motivated by a special property of the $c$-transform of 1-Lipschitz functions, when $c = d$ for any metric $d$: if $f$ is 1-Lipschitz, then $f^c = -f$ (Villani, 2008, Sec. 5). It can also be shown, that a Kantorovich potential $\varphi^*$ solving the dual problem (6) is 1-Lipschitz when $c = d$, and therefore the WGAN minimax objective can be written as

$$\min_{\omega'} \max_\omega \left\{ \mathbb{E}_{x\sim\mu_t} \left[\varphi_\omega(x)\right] - \mathbb{E}_{z\sim\mu_s} \left[\varphi_\omega(g_{\omega'}(z))\right] \right\}. \tag{17}$$

In the WGAN case, there is no restriction on the range of $\varphi_\omega$ as opposed to the GAN case above. The assumptions above require enforcing $\varphi_\omega$ to be 1-Lipschitz. This poses a main implementational difficulty in the WGAN formulation, and has been subjected to a considerable amount of research.

In this work, we will investigate the original approach by weight clipping (Arjovsky et al., 2017) and the popular approach by gradient norm penalties for the discriminator (Gulrajani et al., 2017). We furthermore consider a more direct approach that computes the $c$-transform over minibatches (Mallasto et al., 2019), avoiding the need to ensure Lipschitzness. We also discuss an entropic relaxation approach through $(c, \epsilon)$-transforms over minibatches, introduced by Genevay et al. (2016). In the original work, the discriminator $\varphi_\omega$ is expressed as a sum of kernel functions, however, in this work we will consider multi-layer perceptrons (MLPs), as we do with the other methods we consider.

### 3.1 ESTIMATING THE 1-WASSERSTEIN METRIC

In the experimental section, we will consider four ways to estimate the 1-Wasserstein distance between two measures $\mu$ and $\nu$, these being the weight clipping (WC), gradient penalty (GP), $c$-transform and

$(c, \epsilon)$-transform methods. To this end, we now discuss how these estimates are computed in practice by sampling minibatches of size $N$ from $\mu$ and $\nu$, yielding $\{x_i\}_{i=1}^N$ and $\{y_i\}_{i=1}^N$, respectively, at each training iteration. Then, with each method, the distance is estimated by maximizing a model specific expression that relates to the dual formulation of the 1-Wasserstein distance in (6) over the minibatches. In practice, this maximization is carried out via gradient ascent or one of its variants, such as Adam (Kingma & Ba, 2014) or RMSprop (Tieleman & Hinton, 2012).

**Weight clipping (WC).** The vanilla WGAN enforces $K$-Lipschitzness of the discriminator at each iteration by forcing the weights $W^k$ of the neural network to lie inside some box $-\xi \leq W^k \leq \xi$, considered coordinate-wise, for some small $\xi > 0$ ($\xi = 0.01$ in the original work). Here $k$ stands for the $k^{\text{th}}$ layer in the neural network. Then, the identity for the $c$-transform (with $c = d$) of 1-Lipschitz maps is used, and so (17) can be written as

$$\max_\omega \left\{ \frac{1}{N} \sum_{i=1}^N \varphi_\omega(x_i) - \frac{1}{N} \sum_{i=1}^N \varphi_\omega(y_i) \right\}. \tag{18}$$

**Gradient penalty (GP).** The weight clipping is omitted in WGAN-GP, by noticing that the 1-Lipschitz condition implies that $\|\nabla_x \varphi_\omega(x)\| \leq 1$ holds for $x$ almost surely under $\mu$ and $\nu$. This condition can be enforced through the penalization term $\mathbb{E}_{x \sim \chi}\left[ \max\left(0, 1 - \|\nabla_x \varphi_\omega(x)\|\right)^2 \right]$, where $\chi$ is some reference measure, proposed to be the uniform distribution between pairs of points of the minibatches by Gulrajani et al. (2017). The authors remarked that in practice it suffices to enforce $\|\nabla_x \varphi_\omega(x)\| = 1$, and thus the objective can be written as

$$\max_\omega \left\{ \frac{1}{N} \sum_{i=1}^N \varphi_\omega(x_i) - \frac{1}{N} \sum_{i=1}^N \varphi_\omega(y_i) - \frac{\lambda}{M} \sum_{i=1}^M \left(1 - \|\nabla_{z=z_i} \varphi_\omega(z)\|\right)^2 \right\}, \tag{19}$$

where $\lambda$ is the magnitude of the penalization, which was chosen to be $\lambda = 10$ in the original paper, and $\{z_i\}_{i=1}^M$ are samples from $\chi$.

**$c$-transform.** Enforcing 1-Lipschitzness has the benefit of reducing the computational cost of the $c$-transform, but the enforcement introduces an additional cost, which in the gradient penalty case is substantial. The $\text{ADM}(c)$ constraint can be taken into account directly, as done in $(q, p)$-WGANs (Mallasto et al., 2019), by directly computing the $c$-transform over the minibatches as

$$\varphi_\omega^c(y_i) \approx \widehat{\varphi_\omega^c}(y_i) = \min_j \left\{ c(x_j, y_i) - \varphi_\omega(x_j) \right\}, \tag{20}$$

where $c = d_2$ in the 1-Wasserstein case. This amounts to the relatively cheap operation of computing the row minima of the matrix $A_{ij} = c(x_j, y_i) - \varphi_\omega(x_j)$. The original paper proposes to include penalization terms to enforce the discriminator constraints, however, this is unnecessary as the $c$-transform enforces the constraints. Therefore, the objective can be written as

$$\max_\omega \left\{ \frac{1}{N} \sum_{i=1}^N \varphi_\omega(x_i) + \frac{1}{N} \sum_{i=1}^N \widehat{\varphi_\omega^c}(y_i) \right\}. \tag{21}$$

**$(c, \epsilon)$-transform.** As discussed in Section 2, entropic relaxation applied to $W_1$ results in the $(1, \epsilon)$-Sinkhorn divergence $S_1^\epsilon$, which satisfies $S_1^\epsilon \to W_1$ as $\epsilon \to 0$. Then, $S_1^\epsilon$ can be viewed as a smooth approximation to $W_1$. The benefits of this approach are that $\varphi_\omega$ is not required to satisfy the $\text{ADM}(c)$ constraint, and the resulting transport plan is smoother, providing robustness towards noisy samples.

The expression (13) for $S_1^\epsilon$ consists of three terms, where each results from solving an entropy relaxed optimal transport problem. As stated by Feydy et al. (2018, Sec. 3.1), the terms $\text{OT}_1^\epsilon(\mu, \mu)$ and $\text{OT}_1^\epsilon(\nu, \nu)$ are straight-forward to compute, and tend to converge within couple of iterations of the symmetric Sinkhorn-Knopp algorithm. For efficiency, we approximate these terms with one Sinkhorn-Knopp iteration. The discriminator is employed in approximating $\text{OT}_1^\epsilon(\mu, \nu)$, which is done by computing the $(c, \epsilon)$-transform defined in (11) over the minibatches

$$\varphi_\omega^c(y_i) \approx \widehat{\varphi_\omega^{(c, \epsilon)}}(y_i) = -\epsilon \log \left( \frac{1}{N} \sum_{j=1}^N \exp \left( -\frac{1}{\epsilon} \left(\varphi_\omega(x_j) - c(x_j, y_i)\right) \right) \right), \tag{22}$$

and so we write the objective (12) for the discriminator as

$$\max_{\omega} \left\{ \frac{1}{N} \sum_{i=1}^{N} \varphi_{\omega}(x_i) + \frac{1}{N} \sum_{j=1}^{N} \widehat{\varphi_{\omega}^{(c,\epsilon)}}(y_j) \right\}. \tag{23}$$

## 4 EXPERIMENTS

We now study how efficiently the four methods presented in Section 3.1 estimate the 1-Wasserstein metric. The experiments use the MNIST (LeCun et al., 1998), CIFAR-10 (Krizhevsky et al., 2009), and CelebA (Liu et al., 2015) datasets. On these datasets, we focus on two tasks: approximation and stability. By approximation we mean how well the *minibatch-wise* distance between two measures can be computed, and by stability how well these minibatch-wise distances relate to the 1-Wasserstein distance between the two full measures.

**Implementation.** In the approximation task, we model the discriminator as either (i) a simple multilayer perceptron (MLP) with two hidden layers of width 128 and ReLU activations, and a linear output, or (ii) a convolutional neural network architecture (CNN) used in DCGAN (Radford et al., 2015). In the stability task we use the simpler MLPs for computational efficiency. The discriminator is trained by optimizing the objective function using stochastic or batch gradient ascent. For the gradient penalty method, we use the Adam optimizer with learning rate $10^{-4}$ and beta values $(0, 0.9)$. For weight clipping we use RMSprop

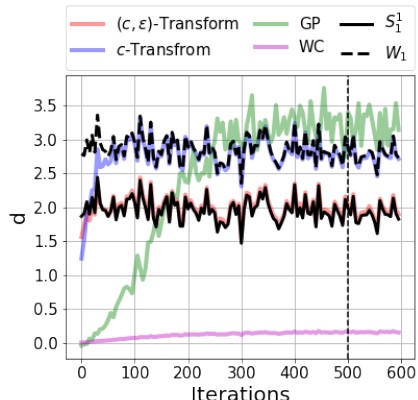

Figure 1: Estimating the distance between two standard 2-dimensional Gaussian distributions that have been shifted by $\pm[1, 1]$.

with learning rate $5 \times 10^{-5}$ as specified in the original paper by Arjovsky et al. (2017). Finally, for the $c$-transform and the $(c, \epsilon)$-transform, we use RMSprop with learning rate $10^{-4}$.

The estimated distances $d_{\text{est}}$ obtained from the optimization are compared to ground truth values $d_{\text{ground}}$ computed by POT[1]. The $(c, \epsilon)$-transform might improve on the POT estimates when $d_{\text{est}} > d_{\text{ground}}$, as both values result from maximizing the same unconstrained quantity. This discrepancy cannot be viewed as error, which we quantify as

$$\text{err}(d_{\text{est}}, d_{\text{ground}}) = \max(0, d_{\text{ground}} - d_{\text{est}}). \tag{24}$$

Note that this is a subjective error based on the POT estimate that can also err, and not absolute error. In practice POT is rather accurate, and we found this to make only a small difference (see Figure 2 with the ground truth and estimated distances visualized). The weight clipping and the gradient penalty methods might also return a higher value than POT, but in this case it is not guaranteed that the discriminators are admissible, meaning that the constraints of the maximization objective would not be satisfied. In the case of the $c$-transform, the discriminators are always admissible. However, the POT package's *ot.emd* (used to compute the 1-Wasserstein distance ground truth) does not utilize the dual formulation for computing the optimal transport distance, and therefore we cannot argue in the same way as in the $(c, \epsilon)$-transform case. As the Sinkhorn divergence (13) consists of three terms that are each maximized, we measure the error as the sum of the errors given in (24) of each term.

**Approximation.** We divide the datasets into two, forming the two measures $\mu$ and $\nu$, between which the distance is approximated, and train the discriminators on 500 *training minibatches* $\mu_k \subset \mu$ and $\nu_k \subset \nu$, $k = 1, ..., 500$, of size 64. See Section 3.1 for how the discriminator objectives. Then, without training the discriminators further, we sample another 100 *evaluation minibatches* $\mu'_l \subset \mu$ and $\nu'_l \subset \nu$, $l = 1, ..., 100$, and use the discriminators to approximate the *minibatch-wise* distance between each $\mu'_l$ and $\nu'_l$. This approximation will then be compared to the ground-truth minibatch-wise distance computed by POT. We run this experiment 20 times, initializing the networks again each time, and report the average error in Table 1. Note that the discriminators are not updated for the last 100 iterations. Results for a toy example between two Gaussians are presented in Fig. 1.

[1]Python Optimal Transport, https://pot.readthedocs.io/.

| **MLP** | MNIST | CIFAR10 | CelebA |
|---|---|---|---|
| WC | $14.98 \pm 0.32$ | $27.26 \pm 0.61$ | $48.65 \pm 1.29$ |
| GP | $14.89 \pm 0.38$ | $27.14 \pm 0.87$ | $48.00 \pm 2.88$ |
| $c$-transform | $0.82 \pm 0.16$ | $1.53 \pm 0.29$ | $2.84 \pm 0.49$ |
| $(c, 0.1)$-transform | $0.43 \pm 0.17$ | $1.29 \pm 0.48$ | $2.52 \pm 1.28$ |
| $(c, 1)$-transform | $(1.12 \pm 4.76) \times 10^{-10}$ | $(0.26 \pm 5.97) \times 10^{-4}$ | $0.04 \pm 0.26$ |
| **ConvNet** | MNIST | CIFAR10 | CelebA |
| WC | $20.73 \pm 18.59$ | $27.28 \pm 0.63$ | $48.72 \pm 1.33$ |
| GP | $14.78 \pm 0.54$ | $25.20 \pm 24.32$ | $96.19 \pm 77.90$ |
| $c$-transform | $0.79 \pm 0.16$ | $1.00 \pm 0.26$ | $2.11 \pm 0.46$ |
| $(c, 0.1)$-transform | $0.42 \pm 0.17$ | $0.60 \pm 0.41$ | $1.74 \pm 1.13$ |
| $(c, 1)$-transform | $(0.23 \pm 1.90) \times 10^{-8}$ | $(0.40 \pm 3.60) \times 10^{-13}$ | $0.02 \pm 0.17$ |

Table 1: **Approximation**. For each method, the discriminators are trained 20 times for 500 iterations on minibatches of size 64 drawn without replacement, after which training is stopped and the error between the ground truth and the estimate are computed.

| MNIST | $(c, 1)$-transform | $c$-transform | GP | WC |
|---|---|---|---|---|
| $N = 512, M = 64$ | $17.20 \pm 0.16$ | $13.87 \pm 0.23$ | $4.25 \pm 0.49$ | $2.10 \pm 0.26$ |
| $N = 512, M = 512$ | $16.95$ | $12.64$ | $4.21$ | $2.03$ |
| $N = 64, M = 64$ | $17.45 \pm 0.06$ | $14.12 \pm 0.13$ | $1.54 \pm 0.25$ | $1.12 \pm 0.13$ |
| $N = 64, M = 512$ | $16.76$ | $11.4$ | $1.49$ | $1.08$ |
| Ground truth | $14.22$ | $12.65$ | $12.65$ | $12.65$ |
| CIFAR10 | $(c, 1)$-transform | $c$-transform | GP | WC |
| $N = 512, M = 64$ | $29.98 \pm 0.28$ | $26.44 \pm 0.25$ | $11.04 \pm 1.16$ | $3.85 \pm 0.67$ |
| $N = 512, M = 512$ | $29.41$ | $24.77$ | $11.10$ | $4.00$ |
| $N = 64, M = 64$ | $29.67 \pm 0.41$ | $26.21 \pm 0.40$ | $3.25 \pm 0.53$ | $2.19 \pm 0.23$ |
| $N = 64, M = 512$ | $29.18$ | $24.16$ | $3.59$ | $2.34$ |
| Ground truth | $26.10$ | $24.78$ | $24.78$ | $24.78$ |
| CelebA | $(c, 1)$-transform | $c$-transform | GP | WC |
| $N = 512, M = 64$ | $50.55 \pm 0.86$ | $46.56 \pm 0.89$ | $28.07 \pm 10.61$ | $19.18 \pm 73.86$ |
| $N = 512, M = 512$ | $48.42$ | $43.06$ | $28.17$ | $20.93$ |
| $N = 64, M = 64$ | $50.80 \pm 0.91$ | $46.83 \pm 0.86$ | $10.24 \pm 7.31$ | $13.98 \pm 39.54$ |
| $N = 64, M = 512$ | $47.60$ | $41.80$ | $10.10$ | $15.20$ |
| Ground truth | $43.74$ | $43.07$ | $43.07$ | $43.07$ |

Table 2: **Stability**. The discriminators are trained using two training batch sizes, $N = 64$ and $N = 512$. Then, the distances between the measures are estimated, by evaluating the discriminators on the full measures (of size $M = 512$), or by evaluating minibatch-wise with batch size $M = 64$. Presented here are the distances approximated by each way of training the discriminators.

As Table 1 shows, the $c$-transform approximates the minibatch-wise 1-Wasserstein distance far better than weight clipping or gradient penalty, and $(c, \epsilon)$-transform does even better at approximating the 1-Sinkhorn divergence. The low errors in this case are due to the $(c, \epsilon)$-transform outperforming the POT library, which results in an error of 0 on many iterations, which also explains why the error variance is so high in the $\epsilon = 1$ case.

**Stability.** We train the discriminators for 500 iterations on small datasets of size 512, that form subsets of the datasets mentioned above. We train with two minibatch sizes $N = 64$ and $N = 512$. We then compare how the resulting discriminators estimate the minibatch-wise and total distances, that is, the evaluation minibatches are of size $M = 64$ and $M = 512$. Letting $\Psi_{\text{Method}}$ be the objective presented in 3.1 for a given method, we train the discriminator by maximizing $\Psi_{\text{Method}}(\phi, \mu_N, \nu_N)$, and finally compare $\Psi_{\text{Method}}(\phi, \mu_M, \nu_M)$ for different $M$. The results are presented in Table 2. An experiment on CIFAR10 was carried out to illustrate how the distance estimate between the full measures behaves when trained minibatch-wise, which is included in Appendix A.

The ground truth values computed using POT are also included, but are not of the main interest in this experiment. The focus is on observing how different batch-sizes on training and evaluation affect the resulting distance. For the $c$-transform and $(c, \epsilon)$-transform, the results varies depending on whether the distances are evaluated minibatch-wise or on the full datasets. On the other hand, for the gradient penalty and weight clip methods, the training batch-size has more effect on the result, but the minibatch-wise and full evaluations are comparable.

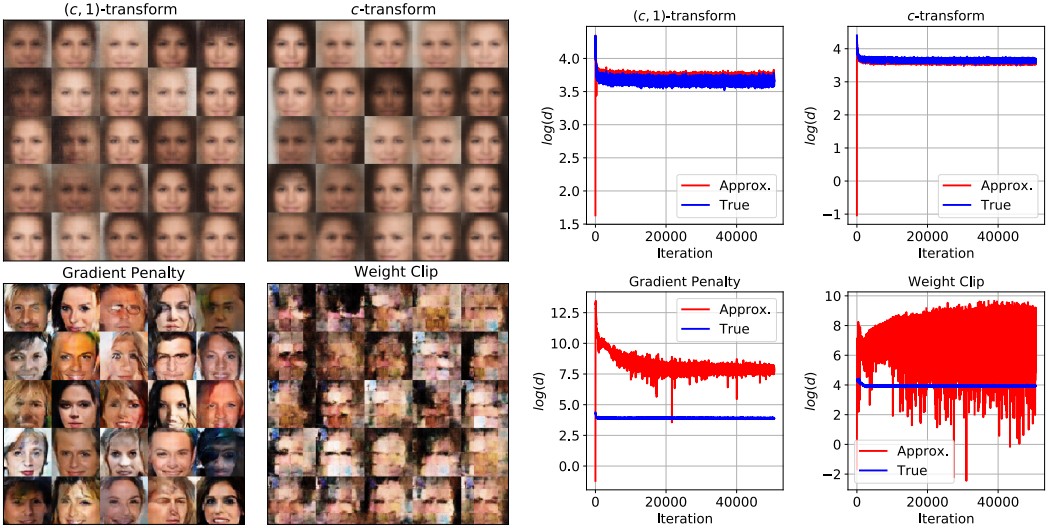

Figure 2: Approximating the minibatch-wise distances while training a generator. Left: generated faces after $5 \times 10^4$ generator iterations. Right: True and estimated log-distances between sampled batches. The large values of GP and WC can be attributed to the fact that they are susceptible to failure in enforcing the Lipschitz constraints on the discriminator.

**WGAN training.** Finally, we measure how the different methods fare when training a Wasserstein GAN. For this, we use the architecture from DCGAN (Radford et al., 2015) to learn a generative model on the CelebA dataset. During this training, we compute the POT ground truth distance between presented minibatches, and compare these to the estimated distances given by the discriminators. This is carried out for a total of $5 \times 10^4$ generator iterations, each of which is accompanied by 5 discriminator iterations. For $(c, \epsilon)$- and $c$-transform the discriminators are evaluated on the fake samples, which gave similar results to evaluating on the real samples. The results are presented in Fig. 2. The results clearly show how the $(c, \epsilon)$-transform and $c$-transform estimate the Wasserstein distances at each iteration better than gradient penalty or weight clipping. However, the resulting images are blurry and look like averages of clusters of faces. The best quality images are produced by the gradient penalty method, whereas weight clipping does not yet converge to anything meaningful. We include the same experiment ran with the simpler MLP architecture for the discriminator, while the generator still is based on DCAN, in Appendix B.

## 5 DISCUSSION

Based on the experiments, $(c, \epsilon)$-transform and $c$-transform are more accurate at computing the minibatch distance and estimating the batch distance than the gradient penalty and weight clipping methods. However, despite the lower performance of the latter methods, in the generative setting they produce more unique and compelling samples than the former. This raises the question, whether the exact 1-Wasserstein distance between batches is the quantity that should be considered in generative modelling, or not. On the other hand, an interesting direction is to study regularization strategies in conjunction with the $(c, \epsilon)$- and $c$-transforms to improve generative modelling with less training.

The results of Table 2 indicate that the entropic relaxation provides stability under different training schemes, endorsing theoretical results implying more favorable sample complexity in the entropic case. In contrast to what one could hypothetise, the blurriness in Fig. 2 seems not to be produced by the entropic relaxation, but the $c$-transform scheme.

Finally, it is interesting to see how the gradient penalty method performs so well in the generative setting, when based on our experiments, it is not able to approximate the 1-Wasserstein distance so well. What is it, then, that makes it such a good objective in the generative case?

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

## A    APPENDIX: STABILITY DURING TRAINING

Related to the **Stability** experiment, we train discriminators modelled by the simple MLPs (see Section 4) to approximate the distance between two measures $\mu$ and $\nu$, which both are subsets of the CIFAR10 dataset of size 512. We train for 15000 iterations with training minibatch size of 64, and report the estimated minibatch-wise distances with the estimated distances between the full measures in Fig. 3.

The experiments demonstrate, how $(c, \epsilon)$-transform and $c$-transform converge rapidly compared to gradient penalty and weight clipping method, which have not entirely converged after 15000 iterations. Also visible is the bias resulting form minibatch-wise computing of the distances compared to the distance between the full measures.

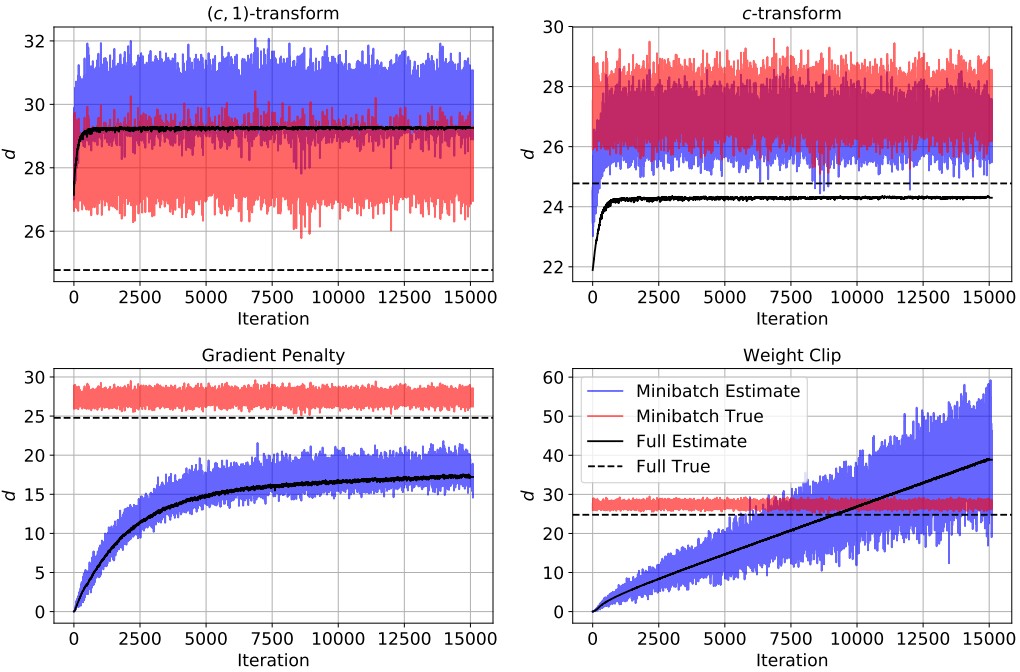

Figure 3: **Stability**. While training the discriminators on minibatches, taken from two measures consisting of 512 samples from CIFAR10, we also report the estimated distance between the full measures.

## B  APPENDIX: WGAN TRAINING WITH MLP

We repeat the **WGAN training** experiment with the simpler MLP architecture (see Section 4) for the discriminators. The distance estimates at each iteration are given in Fig. 4, and generated samples in Fig. 5.

The training process for $(c, \epsilon)$- and $c$-transforms is more unstable with the MLP, as notable in the sudden jumps in the true distance between minibatches. This seems to be caused when the discriminator underestimates the distance. The fluctuation between estimated distances is much higher in the gradient penalty and weight clipping cases, but the decrease in the true distance between minibatches is still consistent. Notice how the fluctuation decreases considerably when we use a ConvNet architecture for the gradient penalty method in Fig. 2.

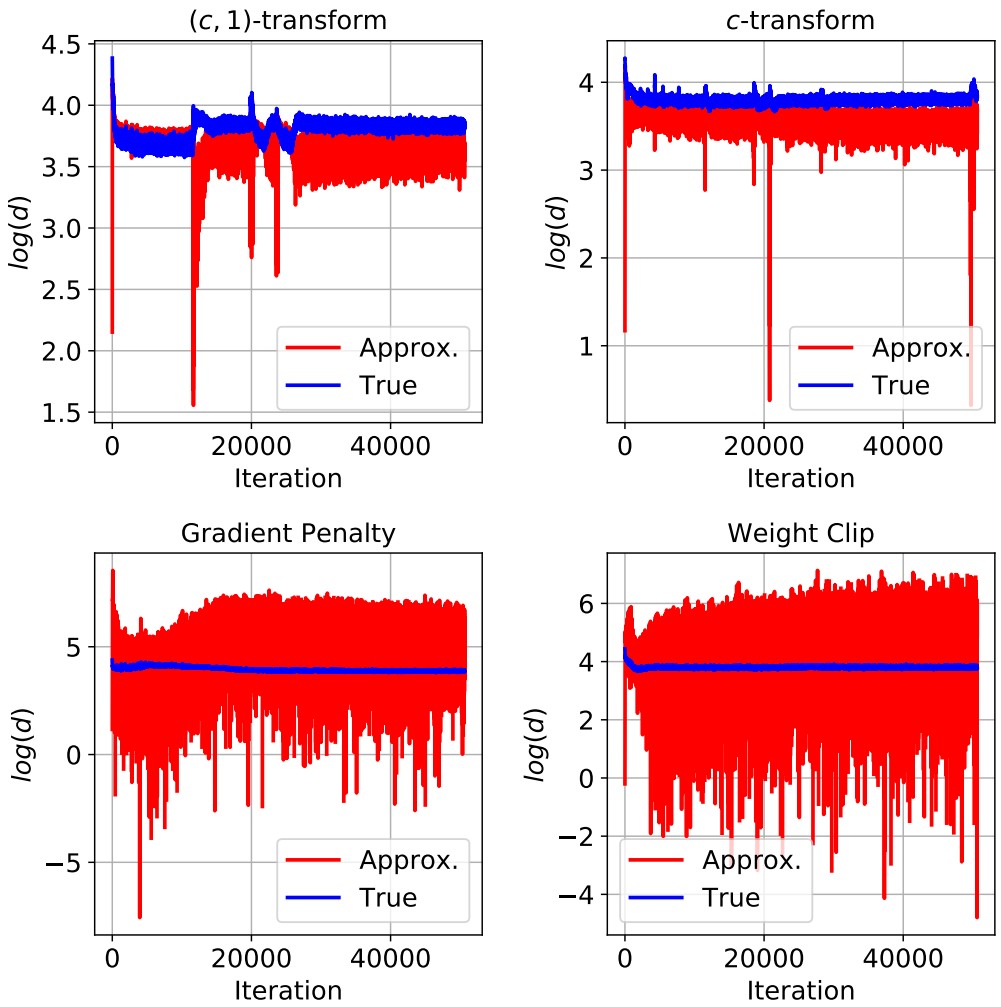

Figure 4: Repeating the experiment presented in Fig. 2, but with the simpler MLP architecture for the discriminator. Presented here are the estimated batchwise distances at each iteration against the ground truth.

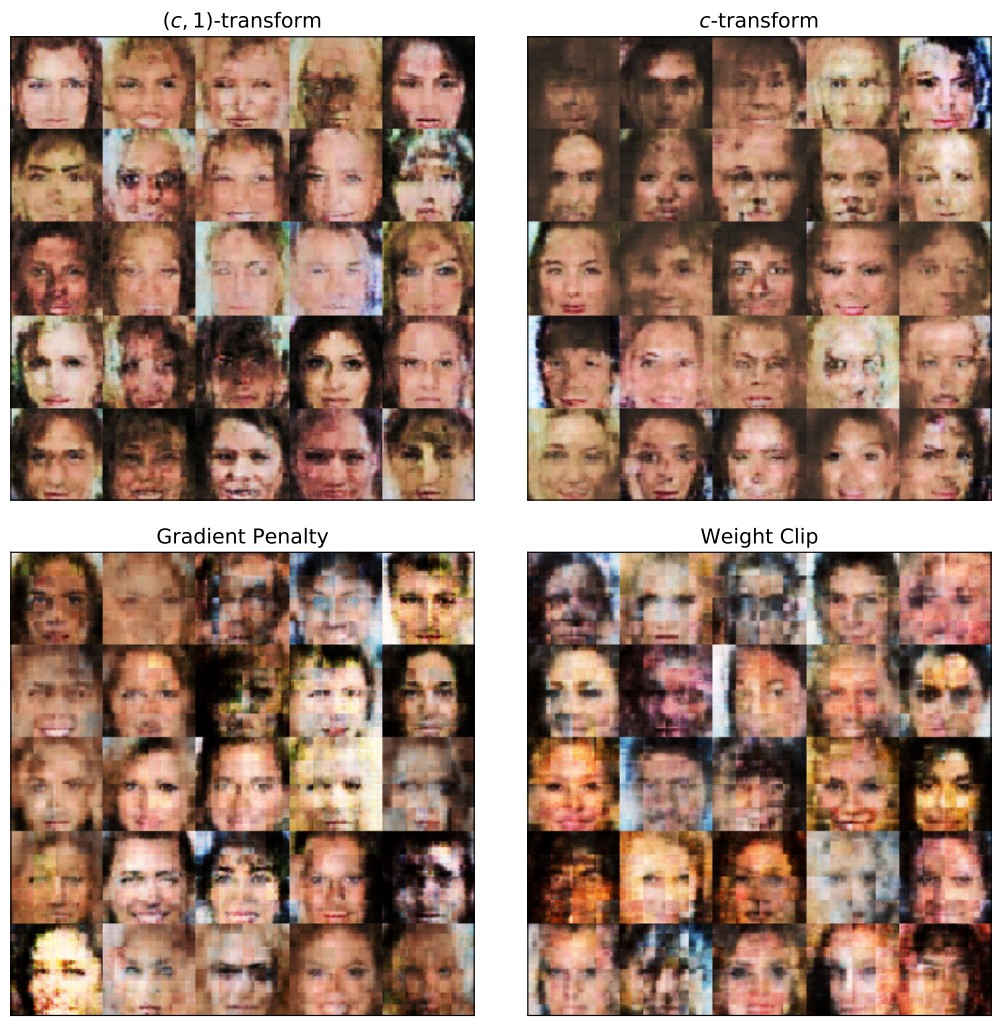

Figure 5: Repeating the experiment presented in Fig. 2, but with the simpler MLP architecture for the discriminator. Presented here are generated samples after $5 \times 10^4$ iterations.

