# OpenReview forum: "How Well Do WGANs Estimate the Wasserstein Metric?"
_ICLR.cc/2020/Conference — Reject_

### Official Review · AnonReviewer1 · 2019-10-22
**Official Blind Review #1**

**Rating:** 1

**Review:**

This paper has studied the efficiency and stability of computing the Wasserstein metric through its dual formulation under weight clipping, gradient penalty, c-transform and (c- ϵ)-transform. The results show that (c- ϵ)-transform and c-transform give more estimation of the Wasserstein distances than the gradient penalty and weight clipping methods in the given experiments, but the gradient penalty method produces more compelling samples in the generative setting.

The paper is well written and the experiment section is extensive. However, it is more like an extended experiment report to me which is very valuable but lacks sufficient technical novelty expected at ICLR.

Another comment is that when the authors mentioned "... are compared to ground truth values d_ground computed by POT", there needs more explanations on what that library actually does to compute the Wasserstein distance to make the paper self-contained, e.g. what exact algorithms it uses as there are also dozens of different algorithms implemented in that library.

In Section 3.1, the authors state that "it tends to converge within couple of iterations of the symmetric Sinkhorn-Knopp algorithm. For efficiency, we approximate these terms with one Sinkhorn-Knopp iteration". What is the extent of sacrifice in accuracy due to this approximation? The authors should provide more evidences to justify the approximation.

**Experience Assessment:**

I have read many papers in this area.

**Review Assessment: Checking Correctness Of Derivations And Theory:**

I assessed the sensibility of the derivations and theory.

**Review Assessment: Checking Correctness Of Experiments:**

I assessed the sensibility of the experiments.

**Review Assessment: Thoroughness In Paper Reading:**

I read the paper at least twice and used my best judgement in assessing the paper.

---

> ### Author Response · Authors · 2019-11-11
> **Author feedback**
>
> Thank you for pointing out where we could improve. Below, we will address those comments, for which the explanations will also be added to the paper. Furthermore, we give arguments to why we view the contribution as a fit for ICLR.
>
> We utilize POT’s ot.emd method to compute the ordinary optimal transport quantity, which is implemented using the method in [1]. We apologize for not explaining which method is used in the entropic case, where the ot.sinkhorn method is utilized with the ‘sinkhorn’ option.
>
> The sacrifice due to only applying one Sinkhorn-Knopp iteration for the unbiasing terms is slight, as can be seen from the resulting errors with respect to the ground truth in the experiments.
>
>
> We view the main contributions of this paper to be the study of the approximation and estimation (stability) of the WGAN methods, which unfortunately has not been considered to its rightful extend in the literature yet. On the technical side, we are the first to consider the smooth c-transform in the WGAN setting, although it relates closely to [2], where the primal formulation of entropic optimal transport is considered. We are happy to hear that you consider the experimentation extensive. On top of new methodological contributions, it is important to also consider comparative studies to guide the development of new methodologies. For example, it is important to consider which functional classes the discriminator should belong to, which reflects on the representation part of ICLR. From our experiments, it seems that, when evaluating optimal transport quantities in the general setting (independent of generation), the $c$-transform provides a meaningful class. In the generative setting the task gets more complicated due to the minimax nature of the game.
>
> [1] Bonneel, N., Van De Panne, M., Paris, S., & Heidrich, W. (2011, December). Displacement interpolation using Lagrangian mass transport. In ACM Transactions on Graphics (TOG) (Vol. 30, No. 6, p. 158). ACM.
>
> [2] Genevay, A., Peyre, G. & Cuturi, M.. (2018). Learning Generative Models with Sinkhorn Divergences. Proceedings of the Twenty-Fi

---

### Official Review · AnonReviewer2 · 2019-10-23
**Official Blind Review #2**

**Rating:** 3

**Review:**


[Summary]
This paper provides an empirical evaluation of commonly used discriminator training strategies in estimating the Wasserstein distance between distributions. The paper finds that methods motivated from optimal transport theory, e.g. c-transform and (c,\eps)-transform, perform better in evaluating the Wasserstein distance than methods commonly used in WGAN practice such as weight clipping and gradient penalty. However, when deployed in WGANs as the discriminator training strategy, these methods do not generate images as high-quality as the gradient penalty method.

[Pros]
The question considered in this paper, i.e. how well does various discriminator strategies (as proxies of the infeasible all 1-Lipschitz discriminator) perform for *evaluating* the Wasserstein distributions, is important for strengthening our understanding of generative models. The main result that methods that are better at computing W may not be better at generating images is interesting, and agrees with the theoretical insight (e.g. Arora et al. 2017) that *non-parametric* minimization of W may not be a good explanation for the generative power of WGANs.

[Cons]
I have concerns about the specific setting of “mini-batch distance” considered in this paper, in that whether it is really a sensible task (and can say anything about computing W) given the curse of dimensionality of estimating W from samples. The paper did not really discuss this issue, and from my own thoughts I don’t think the task avoids this issue.

From my understanding, the “Approximation” experiment does the following:
(1) Set (\mu, \nu) to be a random split of a dataset and consider them to be the populations. Let’s let f_\star denote the (ground truth) optimal discriminator between (\mu, \nu).
(2) Train discriminators (f_wc, f_gp, f_c, f_ceps) (using the different algorithms) from training batches from (\mu, \nu).
(3) Evaluate <f, \mu’_l - \nu’_l> where (\mu’_l, \nu’_l) are fresh test batches from (\mu, \nu) and f is one of the above trained discriminators.
In comparison, the “ground truth” computes W(\mu’_l, \nu’_l) = <f_l, \mu’_l, \nu’_l> from the POT package (though not necessarily through explicitly computing f_l.)

The issue with this is that we expect the method to perform well if f ~= f_l, which can be achievable if all the f_l’s are similar (and hopefully they’re all approximately equal to f_\star.) However I don’t think this is true -- as (\mu’_l, \nu’_l) are samples, and because of the curse of dimensionality, we should expect the f_l’s to be quite different from each other. (Otherwise if they’re really just ~= f_\star, then we can use standard concentration to show the W(\mu’_l, \nu’_l) ~= W(\mu, \nu), which we know is not true from curse of dim.)

Given this, I don’t think the task of comparing <f, \mu’_l, \nu’_l> with the POT results really says anything about their power in computing W. I would be glad though to hear back from the authors to see if my understanding is accurate, and adjust my evaluation from there.


**Experience Assessment:**

I have published one or two papers in this area.

**Review Assessment: Checking Correctness Of Derivations And Theory:**

I assessed the sensibility of the derivations and theory.

**Review Assessment: Checking Correctness Of Experiments:**

I carefully checked the experiments.

**Review Assessment: Thoroughness In Paper Reading:**

I read the paper at least twice and used my best judgement in assessing the paper.

---

> ### Author Response · Authors · 2019-11-11
> **Author feedback**
>
> Thank you for the comments, we especially appreciate your point on whether evaluating the 'batchwise' distance is meaningful, as this is also something we had to spend some time thinking on. We will address the comments below.
>
> Your criticism on the approximation example is point on, which is exactly why we also considered the stability experiment, where we compared the ‘batchwise’ and ‘full’ evaluations, i.e., <f_\star, \mu-\nu>, <f_star, \mu_I-\nu_I>, <f,\mu-\nu> and <f,\mu_I-\nu_I>. Even in these cases, the weight-clipping and gradient penalty methods are far off. On the other hand, there are successful ‘batchwise’ training algorithms for GANs, such as the Sinkhorn based[1], and for example GAN with quadratic transport [2], which motivates the study of the batchwise approximation.
>
>
> Based on the current methodology, it seems that for general cost functions, the ‘batchwise’ evaluation is the way to go, as is supported by the theoretical results (i.e. the max in the Kantorovich duality can be achieved by the pair (f,f^c)). Additionally, including the Kantorovich potentials in other cases than the fortunate 1-Wasserstein case becomes highly non-trivial. As examples of applications with more general cost functions, we would like to point out an example in quantum chemistry, where the Coulomb cost (related to Coulomb potential) is natural [3]. On the other hand, maximum-likelihood deconvolutions with respect to different noise distributions relates to the minimization of entropic optimal transport with different cost functions[4].
>
> [1] Genevay, A., Peyre, G. & Cuturi, M.. (2018). Learning Generative Models with Sinkhorn Divergences. Proceedings of the Twenty-First International Conference on Artificial Intelligence and Statistics, in PMLR 84:1608-1617
>
> [2] Liu, H., Gu, X., & Samaras, D. (2019). Wasserstein GAN with Quadratic Transport Cost. In Proceedings of the IEEE International Conference on Computer Vision (pp. 4832-4841).
>
> [3] Cotar, C., Friesecke, G., & Klüppelberg, C. (2013). Density functional theory and optimal transportation with Coulomb cost. Communications on Pure and Applied Mathematics, 66(4), 548-599.
>
> [4] Rigollet, P., & Weed, J. (2018). Entropic optimal transport is maximum-likelihood deconvolution. Comptes Rendus Mathematique, 356(11-12), 1228-1235.

---

### Official Review · AnonReviewer3 · 2019-10-23
**Official Blind Review #3**

**Rating:** 3

**Review:**

The paper empirically evaluates different variants of WGAN (with weight clipping, gradient penalty, c-transform, and the generalized c-transform under entropy relaxation). The experiments, mainly performed over three datasets (MNIST, CIFAR10, CelebA), are designed to evaluate how well the Wasserstein distance is approximated, how much these approximations depend on batch sizes, and how good are the obtained generative models.

I find the presentation of the different background work and models to be excellent, especially for someone who's not expert on WGANs like me. However, they may want to check the writing, like the sentence just after (17) or the penalization term between (18) and (19).

The contributions of the paper are experimental. The authors argue that they obtain a surprising observation, which is that "the method best approximating the Wasserstein distance does not produce the best looking images in the generative setting ".
However, the goodness of the approximation is measured with (24), which the authors called "subjective error". I think the authors may want to comment more on this measure, which seems to favor the different transforms.
Also, the quality of the generative models seems to strongly depend on the architectures used in WGAN. The authors' conclusions are based on DCGAN. However, the results obtained with simple MLP and presented in the appendix have not the same clear distinction as with DCGAN.

Overall, although I liked the presentation very much, I feel the experimental results may be a bit too light for a publication in a venue such as ICLR.

**Experience Assessment:**

I do not know much about this area.

**Review Assessment: Checking Correctness Of Derivations And Theory:**

N/A

**Review Assessment: Checking Correctness Of Experiments:**

I assessed the sensibility of the experiments.

**Review Assessment: Thoroughness In Paper Reading:**

I read the paper at least twice and used my best judgement in assessing the paper.

---

> ### Author Response · Authors · 2019-11-11
> **Author feedback**
>
> Thank you for the valuable comments, which we will address below.
>
> Sentence after (17): The range of the discriminator in the WGAN setting can attain any value, and thus its range is not constrained to lie inside of [0,1], unlike in the vanilla GAN case, where the attained value reflects on a probability of a picture being from the given dataset.
>
> Penalization term: Thank you for pointing out the mistake on the discussion about the penalization term. Although 1-Lipschitzness does imply that the gradient norm should lie inside of [0,1], due to the properties of the optimal matching, on the support of the two measures considered the norm should equal to one.
>
>  Subjective error: The error given in (24) only applies to the smooth c-transform case. Therefore, it does give a bias towards the smooth c-transform, but not the vanilla c-transform. The error was described as subjective, and not absolute, as the POT values can err, too.
>
> Dependence on architecture: We agree, that the generative results do seem to depend on the architecture. However, it seems that the key difference is convolutional vs. non-convolutional architectures. We are currently running experiments using the popular resnet architecture, and in the case of the smooth and vanilla c-transforms, the resulting images look very similar to the DCGAN case.
>
> Contribution: We see that the outcomes of the experiments would be valuable to the optimal transport and WGAN communities due to the following aspects: we show that the c-transform methods, which are able to incorporate any cost function, provide a a meaningful functional family for the discriminator. This then brings forth the surprising result we emphasized in the paper, that efficient computation of the optimal transport quantity might be suboptimal for generative purposes.

---

### Decision · Program_Chairs · 2019-12-19

**Decision:**

Reject

**Comment:**

There is insufficient support to recommend accepting this paper.  Generally the reviewers found the technical contribution to be insufficient, and were not sufficiently convinced by the experimental evaluation.  The feedback provided should help the authors improve their paper.